# Structures of the ATP-fueled ClpXP proteolytic machine bound to protein substrate

Xue Fei[1], Tristan A Bell[1], Simon Jenni[2], Benjamin M Stinson[1], Tania A Baker[1,3], Stephen C Harrison[2,3], Robert T Sauer[1]*

[1]Department of Biology, Massachusetts Institute of Technology, Cambridge, United States; [2]Department of Biological Chemistry and Molecular Pharmacology, Harvard Medical School, Boston, United States; [3]Howard Hughes Medical Institute, Chevy Chase, United States

**Abstract** ClpXP is an ATP-dependent protease in which the ClpX AAA+ motor binds, unfolds, and translocates specific protein substrates into the degradation chamber of ClpP. We present cryo-EM studies of the *E. coli* enzyme that show how asymmetric hexameric rings of ClpX bind symmetric heptameric rings of ClpP and interact with protein substrates. Subunits in the ClpX hexamer assume a spiral conformation and interact with two-residue segments of substrate in the axial channel, as observed for other AAA+ proteases and protein-remodeling machines. Strictly sequential models of ATP hydrolysis and a power stroke that moves two residues of the substrate per translocation step have been inferred from these structural features for other AAA+ unfoldases, but biochemical and single-molecule biophysical studies indicate that ClpXP operates by a probabilistic mechanism in which five to eight residues are translocated for each ATP hydrolyzed. We propose structure-based models that could account for the functional results.

*For correspondence:
bobsauer@mit.edu

Competing interests: The authors declare that no competing interests exist.

## Introduction

AAA+ motors harness the energy of ATP hydrolysis to carry out mechanical tasks in cells (*Erzberger and Berger, 2006*). In the ClpXP protease, for example, AAA+ ClpX ring hexamers bind target proteins, unfold them, and translocate the unfolded polypeptide through an axial channel and into the peptidase chamber of ClpP, which consists of two heptameric rings (*Figure 1A*; *Wang et al., 1997*; *Grimaud et al., 1998*; *Ortega et al., 2000*; *Ortega et al., 2002*; *Sauer and Baker, 2011*). In the absence of ClpX or another AAA+ partner, small peptides diffuse into the ClpP chamber through narrow axial pores, but larger peptides and native proteins are excluded and escape degradation (*Grimaud et al., 1998*; *Lee et al., 2010a*). The substrates of *Escherichia coli* ClpXP include aberrant ssrA-tagged proteins, produced by abortive translation, and normal cellular proteins synthesized with degradation tags that program rapid turnover (*Baker and Sauer, 2012*; *Keiler, 2015*).

ClpX subunits consist of a family specific N-terminal domain, which is dispensable for degradation of ssrA-tagged proteins, and large and small AAA+ domains, which contain sequence motifs that mediate ATP binding and hydrolysis, ClpP binding, and substrate recognition (*Figure 1B*; *Baker and Sauer, 2012*). Ring hexamers of ClpX bind the ssrA tag within an axial channel (*Martin et al., 2008a*). Following degron binding, ATP-fueled power strokes pull on and eventually unfold attached native domains (*Kenniston et al., 2003*). Single-chain ClpX$^{\Delta N}$ pseudohexamers, containing six 'subunits' linked by genetically encoded tethers, support ClpP degradation of ssrA-tagged substrates at rates similar to wild-type ClpX (*Martin et al., 2005*). Eliminating ATP hydrolysis in four or five subunits of single-chain pseudohexamers slows but does not prevent ClpP-mediated degradation,

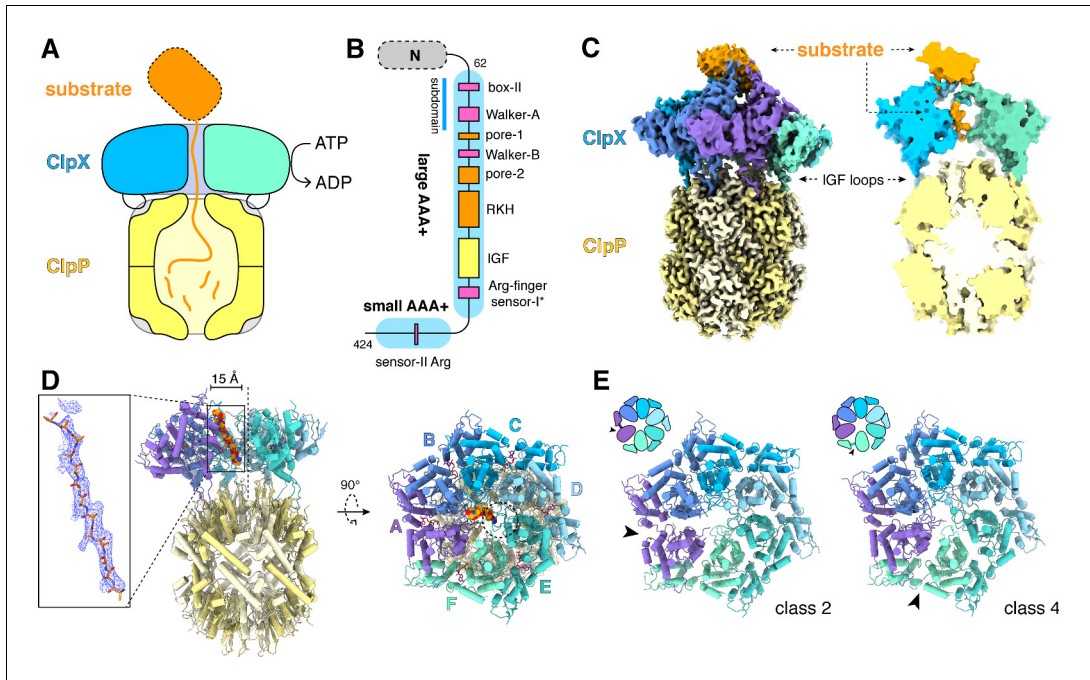

**Figure 1.** ClpXP protease. (**A**) Schematic representation of ClpXP. (**B**) ClpX domain structure and positions of sequence motifs important for ATP binding and hydrolysis (magenta), substrate binding (orange), and ClpP binding (yellow). (**C**) Left. Composite cryo-EM density of ClpX^ΔN/ClpP complex. ClpP is yellow; ClpX^ΔN (class 3) is blue, green, or purple; and substrate is orange. Right. Slice through surface representation of the model showing substrate in the axial channel and the ClpP degradation channel. (**D**) Left. Cartoon representation of the class-4 ClpX^ΔN hexamer (subunit F removed) and its docking with a heptameric ClpP ring. Substrate in the ClpX channel is shown in space-filling representation as a poly-alanine chain; the inset shows substrate as a ball-and-stick model with associated density. The vertical dashed line shows the 7-fold symmetry axis of ClpP. Right. View rotated by 90°. The dashed circle shows the position of the ClpP pore. (**E**) Cartoon representation of class-2 and class-4 ClpX^ΔN hexamers with substrate removed for clarity. Arrows point to the different positions of the seam interface. The online version of this article includes the following figure supplement(s) for figure 1:

**Figure supplement 1.** Cryo-EM data and strategy.

**Figure supplement 2.** Local map resolution and FSC plots.

suggesting that ATP hydrolysis in any one of multiple subunits in the ClpX ring is sufficient to power unfolding and translocation (*Martin et al., 2005*). In optical-trapping experiments using single-chain ClpX and ClpP, the smallest observed translocation steps correspond to movement of five to eight amino acids of the substrate, and kinetic bursts of power strokes produce fast translocation steps two, three, or four-fold larger in terms of the number of residues translocated (*Aubin-Tam et al., 2011*; *Maillard et al., 2011*; *Sen et al., 2013*; *Cordova et al., 2014*; *Olivares et al., 2017*). Such bursts do not occur in repeating patterns, supporting probabilistic but coordinated ATP hydrolysis within the ClpX ring.

We describe here near atomic-resolution single-particle cryo-EM structures of single-chain ClpX pseudohexamers bound to ClpP and protein substrates. These structures show how asymmetric hexameric rings of ClpX dock with symmetric heptameric rings of ClpP and reveal how the pore-1, pore-2 and RKH loops of ClpX function in substrate binding. ClpX adopts a spiral conformation, with neighboring pore-1 loops interacting with every two residues of substrate in the axial channel, as observed in other AAA+ unfolding and remodeling machines (*Puchades et al., 2017*; *de la Peña et al., 2018*; *Dong et al., 2019*; *Majumder et al., 2019*; *Ripstein et al., 2017*; *Gates et al., 2017*; *Zehr et al., 2017*; *Han et al., 2017*; *Monroe et al., 2017*; *Su et al., 2017*; *Sun et al., 2017*; *Yu et al., 2018*; *White et al., 2018*; *Cooney et al., 2019*; *Rizo et al., 2019*; *Shin et al., 2019*; *Twomey et al., 2019*). Based on these structural features, strictly sequential models of ATP hydrolysis and a power stroke that moves two residues of the substrate per translocation step have been

proposed. As noted, however, ClpX does not need to operate in a strictly sequential manner and takes translocation steps substantially longer than two residues. Thus, an apparent incongruity exists between the structural and functional studies. We discuss this conflict and propose structure-based translocation models that reconcile how ClpX might use probabilistic ATP hydrolysis to take larger translocation steps of varying length.

## Results

### Cryo-EM structures

For cryo-EM studies, we used epitope-tagged variants of *Escherichia coli* ClpP and a single-chain variant of *E. coli* ClpX$^{\Delta N}$ with an E185Q mutation to eliminate ATP hydrolysis without compromising nucleotide, ClpP, or substrate binding (*Hersch et al., 2005*; *Martin et al., 2005*). Single-chain ClpX$^{\Delta N}$ was used to ensure subunits of the pseudohexamer do not dissociate during sample preparation. This enzyme has been used previously for many biochemical and single-molecule studies (*Martin et al., 2005*; *Martin et al., 2007*; *Martin et al., 2008a*; *Martin et al., 2008b*; *Aubin-Tam et al., 2011*; *Maillard et al., 2011*; *Glynn et al., 2012*; *Sen et al., 2013*; *Stinson et al., 2013*; *Cordova et al., 2014*; *Iosefson et al., 2015a*; *Iosefson et al., 2015b*; *Olivares et al., 2017*; *Rodriguez-Aliaga et al., 2016*; *Amor et al., 2019*; *Bell et al., 2018*). We purified enzymes separately and incubated ClpX$^{\Delta N}$ (4 µM pseudohexamer), ClpP (2 µM 14-mer), and ATPγS (5 mM) for five min before vitrification. Experiments using ATP instead of ATPγS resulted in fewer ClpXP complexes. Imaging revealed complexes with ClpP bound to one or two ClpX$^{\Delta N}$ hexamers (*Figure 1—figure supplement 1A–C*). We analyzed the more abundant doubly capped complexes. As single-particle data processing with C$_2$ symmetry produced maps with conformational heterogeneity, we used signal-subtraction methods to allow independent classification and refinement of ClpX$^{\Delta N}$ or ClpP density (*Figure 1—figure supplement 1D–F*). We calculated a D$_7$ symmetric map for ClpP and parts of ClpX$^{\Delta N}$ making symmetric contacts, four different classes of symmetry-free ClpX$^{\Delta N}$ maps, and then extended each asymmetric ClpX$^{\Delta N}$ map to include one heptameric ClpP ring. Final structures have good stereochemistry with resolutions from 3.2 to 4.3 Å (*Figure 1—figure supplement 2*; *Supplementary file 1*-Table S1). Substrates were observed in all ClpX$^{\Delta N}$ structures and probably represent bound endogenous peptides/proteins or partially denatured portions of ClpP or ClpX$^{\Delta N}$.

*Figure 1C* shows density for a composite ClpX$^{\Delta N}$/ClpP/substrate complex. The six subunits of ClpX$^{\Delta N}$ formed a shallow spiral (labeled ABCDEF from top to bottom) in all structural classes, which differed largely in substrate density or nucleotide state, and the hexameric ClpX$^{\Delta N}$ and heptameric ClpP rings were slightly offset (*Figure 1D*). The structure of the ClpX$^{\Delta N}$ hexamers in the class-1, class-3, and class-4 EM structures were very similar to each other (pair-wise Cα RMSDs 1.2–1.9 Å). The hexamer in the class-2 structure was generally similar (pair-wise Cα RMSDs 2.6 Å) but differed in the position of the 'seam', a dilated inter-subunit interface that occurs as a result of ring closure. This seam was located between subunits A and B in the class-2 structure and between subunits F and A in the class-1, class-3, and class-4 structures (*Figure 1E*). As discussed below, this difference appears to be related to the identity of the nucleotide bound in subunits A or F.

### ClpX docking with ClpP

Our D$_7$-symmetric map included ClpP$_{14}$ and symmetric interface contacts with ClpX$^{\Delta N}$ (*Figure 2A*). In both heptameric ClpP rings, the N-terminal residues of each subunit formed a collar of β-hairpins creating a pore into the degradation chamber (*Figure 2A–B*, *Figure 2—figure supplement 1A*). ClpP had essentially the same structure in the D$_7$ and symmetry-free ClpXP maps. ClpX IGF loops, named for an Ile$^{268}$-Gly$^{269}$-Phe$^{270}$ sequence, are critical for ClpP binding (*Kim et al., 2001*; *Joshi et al., 2004*; *Martin et al., 2007*; *Amor et al., 2019*) and were responsible for most contacts in our structures (*Figure 2C*). These interactions included packing of the Ile$^{268}$, Phe$^{270}$, and Val$^{274}$ side chains into pockets at interfaces between ClpP subunits (*Figure 2D*). I268L, F270L, and V274A variants have severe ClpP binding defects (*Amor et al., 2019*). Asymmetric ClpX-ClpP docking relies on conformational adjustments in the N- and C-terminal residues of individual IGF loops, allowing the central portion of each IGF loop to contact the flat ClpP ring despite projecting from a spiral (*Figure 2E*). Changes in IGF-loop length decrease ClpP affinity and degradation activity (*Amor et al., 2019*), suggesting that these loops act as shock absorbers to maintain ClpP contacts

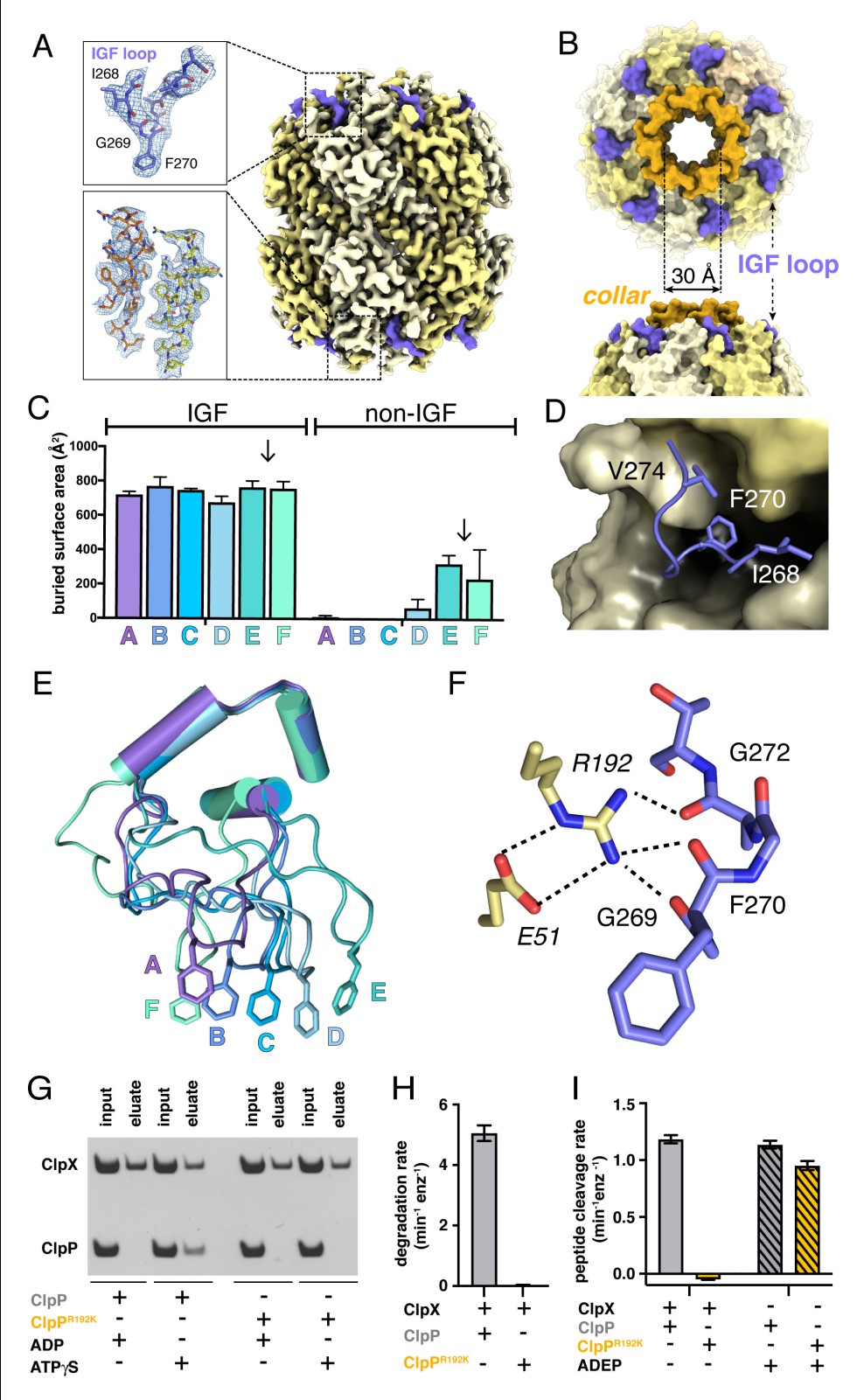

**Figure 2.** ClpP and symmetric IGF loop contacts. (A) Density for ClpP$_{14}$ in the D$_7$ map is shown in yellow. Density for part of the IGF loops of ClpX is shown in blue. The upper inset shows the density as mesh and the fitted model for residues 267–275 of one IGF loop; the lower inset shows density and the fitted model for resides 2–18 of two β-hairpins in the ClpP collar. (B) Top and side views of ClpP showing the axial collar (darker yellow) and pore. (C)

*Figure 2 continued on next page*

*Figure 2 continued*

IGF loops of ClpX make the majority of contacts with ClpP as assessed by buried surface area. Each bar represents a different ClpX$^{\Delta N}$ subunit in the spiral and values are means ± 1 SD for the four classes. The arrows between subunits E and F mark the position of the unoccupied ClpP cleft. (D) The side chains of IGF residues Ile$^{268}$ (I268) Phe$^{270}$ (F270), and Val$^{274}$ (V274) pack into hydrophobic ClpP clefts. (E) After alignment of the large AAA+ cores of the six subunits in the class-4 ClpX$^{\Delta N}$ hexamer, the IGF loops adopt a variety of conformations, helping mediate ClpP binding despite the symmetry mismatch. The side chains of Phe$^{270}$ are shown in stick representation with the IGF loop (residues 263–283) and three flanking helices (residues 254–262 and 284–298) shown in cartoon representation. (F) The side chain of Arg$^{192}$ (R192) in ClpP makes hydrogen bonds with carbonyl oxygens in the IGF loop of ClpX and also forms a salt bridge with Glu$^{51}$ (E51) in a neighboring ClpP subunit. (G) ClpP but not $^{R192K}$ClpP binds ClpX$^{\Delta N}$ in pull-down assays performed in the presence of ATPγS. As a negative control, neither ClpP variant binds ClpX$^{\Delta N}$ in the presence of ADP (*Joshi et al., 2004*). (H) ClpX$^{\Delta N}$ supports degradation of $^{cp7}$GFP-ssrA by ClpP but not by $^{R192K}$ClpP. (I) ClpX$^{\Delta N}$ supports degradation of a decapeptide by ClpP but not by $^{R192K}$ClpP. ADEP-2B activates decapeptide cleavage by both ClpP and $^{R192K}$ClpP. Error bars represent means (n = 3) ± 1 SD.

The online version of this article includes the following figure supplement(s) for figure 2:

**Figure supplement 1.** Structural features of ClpX$^{\Delta N}$/ClpP interface.

during ClpXP machine function. In the symmetry-free maps, the unoccupied pocket in each ClpP heptamer was always located between the pockets bound by IGF loops from ClpX subunits E and F (*Figure 2—figure supplement 1B*; *Video 1*).

In our structures, the side chain of Arg$^{192}$ in ClpP appeared to hydrogen bond to the backbone of the IGF loop (*Figure 2F*). Unlike wild-type ClpP, $^{R192K}$ClpP neither bound ClpX$^{\Delta N}$ in pull-down assays (*Figure 2G*) nor degraded protein substrate in the presence of ClpX$^{\Delta N}$ (*Figure 2H*). Small-molecule acyldepsipeptides (ADEPs) bind in the same ClpP pockets as the ClpX IGF loops and kill bacteria by opening the ClpP pore to facilitate rogue degradation of unstructured proteins (*Brötz-Oesterhelt et al., 2005*; *Kirstein et al., 2009*; *Li et al., 2010*; *Lee et al., 2010b*). ADEPs stimulated R192K and wild-type ClpP decapeptide cleavage to similar extents (*Figure 2I*), establishing that $^{R192K}$ClpP has normal peptidase activity. Thus, Arg$^{192}$ is critical for ClpX binding but not for ADEP binding or ClpP pore opening. Modifying ADEPs to interact with Arg$^{192}$ may increase affinity for ClpP and improve their efficacy as antibiotics.

Crystal structures show that ADEP binding to ClpP results both in opening of the axial pore and

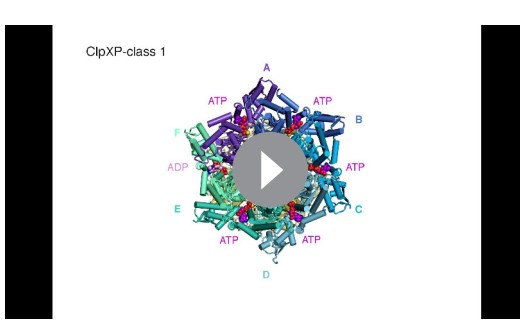

**Video 1.** Top and side views of a ClpX$^{\Delta N}$ hexamer and adjacent ClpP ring for classes 1, 2, 3, and 4. ClpX$^{\Delta N}$ subunits (cartoon representation) are colored blue, green, or purple; carbons in substrate (space-filling representation) are colored orange; carbons in ATP or ADP (space-filling representation) are colored magenta; and ClpP subunits (cartoon representation) are colored yellow with the axial β-hairpin colored gold. Note that substrate above the axial channel of ClpX$^{\Delta N}$ is not shown because the corresponding density was not modeled.

https://elifesciences.org/articles/52774#video1

assembly of its N-terminal sequences into a collar of β-hairpins (*Li et al., 2010*; *Lee et al., 2010b*). Based on a recent cryo-EM structure of *Listeria monocytogenes* ClpXP, it was concluded that ClpX binding does not induce the same widening of the ClpP pore as does ADEP binding (*Gatsogiannis et al., 2019*). By contrast, our results support the opposite conclusion, as the ~30 Å diameter and overall structure of the ClpP pore in our *E. coli* ClpXP structures were extremely similar to the crystal structure of ADEP-activated *E. coli* ClpP (*Li et al., 2010*), with an overall RMSDs of 0.8 Å for all Cα's in a single ClpP ring and 0.6 Å for all Cα's of the seven N-terminal β-hairpins and adjacent α-helices that define the pore diameter.

## Substrate binding

All ClpX$^{\Delta N}$ maps contained substrate density within the axial channel, which we generally modeled as an extended poly-alanine chain (*Figure 1D*), although additional side-chain density of the substrate was modeled as

arginine (residue 5) in the class-1 structure and histidine (residue 3) in the class-3 structure (*Figure 4—figure supplement 1A*). Substrate was built with the N terminus facing ClpP in the class-2 and class-4 structures and in the opposite orientation in the class-1 and class-3 structures. ClpXP can translocate substrates in either the N-to-C or C-to-N direction (*Olivares et al., 2017*). In all structures, the top of the ClpX channel was most constricted by tight packing between ClpX side chains and substrate (*Figure 3A–B*), providing a structural basis for experiments showing that interactions with substrate near the top of the ClpX channel are most important for unfolding grip (*Bell et al., 2019*). The class-1 and class-3 structures also had density corresponding to a roughly globular native domain above the channel (*Figure 1C*, *Figure 4—figure supplement 1B*).

In our structures, the pore-1, pore-2, and RKH loops of ClpX contacted substrate in accord with genetic and biochemical studies (*Siddiqui et al., 2004*; *Farrell et al., 2007*; *Martin et al., 2008a*; *Martin et al., 2008b*; *Iosefson et al., 2015a*; *Iosefson et al., 2015b*; *Rodriguez-Aliaga et al., 2016*). For example, substrate was contacted by the pore-1 loops of subunits B/C/D/E in all classes,

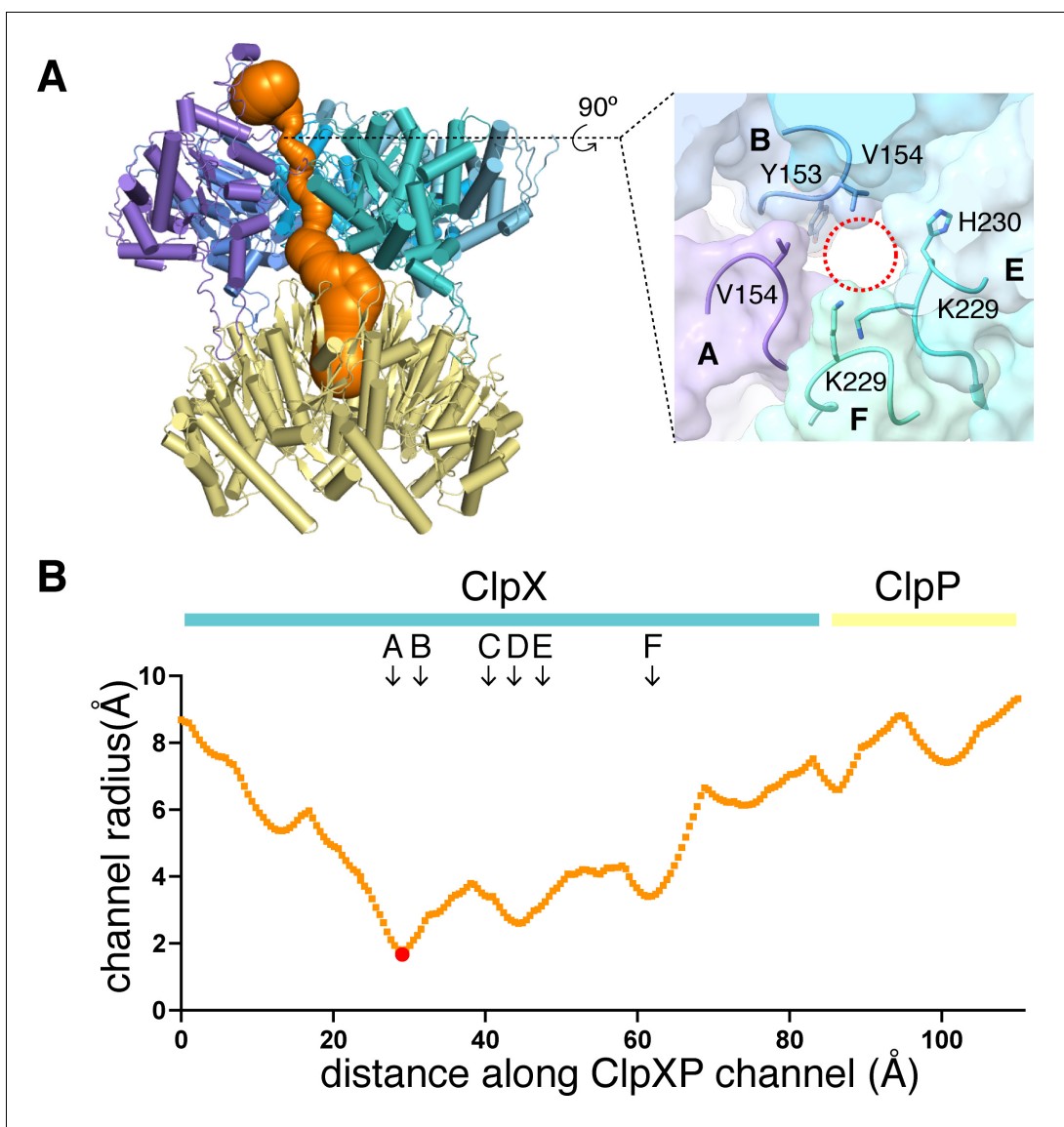

**Figure 3.** Dimensions of axial channel. (**A**) The channel though ClpX and into ClpP (class-1) is most constricted near the top of the ClpX^ΔN hexamer. This panel was made using *Caver* (*Pavelka et al., 2016*). The inset shows that the constriction involves the pore-1 loops of subunits A and B and the RKH loops of subunits E and F. (**B**) Plot of channel radius as a function of channel length. Arrows mark positions of pore-1 loops. Red dot marks the bottleneck of this channel.

by the pore-1 loop of subunit A in class 4, and by the pore-1 loops of subunit F in classes 1 and 2 (*Figure 4A–C*, *Figure 4—figure supplement 1C–D*; *Video 2*). For each engaged pore-1 loop, the side chains of Tyr[153] and Val[154] packed between β-carbons spaced two-residues apart on opposite sides of the extended substrate (*Figure 4A–C*, *Figure 4—figure supplement 1C–D*; *Video 2*). Depending on the class, from three to five pore-2 loops also contacted substrate, with Val[202] making many of these interactions (*Figure 4A, B and D*, *Figure 4—figure supplement 1C–D*; *Video 2*).

The RKH loop, which has not been visualized in previous structures and is unique in ClpX-family enzymes (*Baker and Sauer, 2012*), consisted of an antiparallel β-ribbon stem and a short helix that includes part of the conserved Arg[228]-Lys[229]-His[230] motif (*Figure 4E*). When contacting the globular portion of substrate above the pore, the RKH loops mimicked adjustable structural jacks supporting a house during foundation repair (*Figure 4E*). The RKH loops alter substrate specificity (*Farrell et al., 2007*; *Martin et al., 2008a*), but whether they are critical determinants of substrate binding is unknown. To test this possibility, we constructed and assayed RKH-loop mutants. The most severe mutant (RKH→AAA) did not support ClpP degradation of an ssrA-tagged substrate and hydrolyzed ATP ~5 fold faster than the parent (*Figure 4F–G*). Changing RKH to AKH, KHR, or KRH slowed degradation (*Figure 4F*). A variant [RKH$_2$-AAA]$_2$ pseudohexamer with four wild-type RKH loops and two AAA mutations had a ~3 fold higher $K_M$ for substrate degradation than the parent

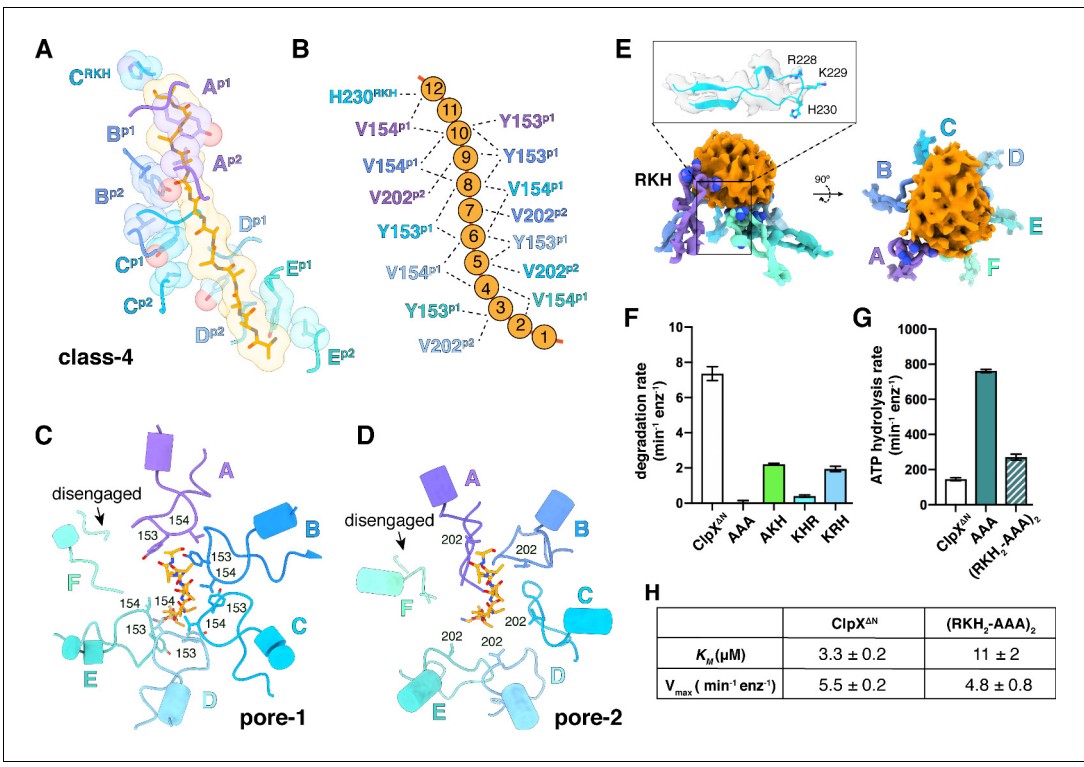

**Figure 4.** Substrate contacts. (**A**) Interactions between substrate (orange, stick and surface representation) and ClpX[ΔN] loops (blue and purple, stick and transparent space-filling representation) in the class-4 structure. Capital letters indicate subunits; superscripts indicate RKH loops, pore-1 (p1) loops, or pore-2 (p2) loops. (**B**) Scheme of interactions shown in panel (**A**). Dashed lines represent distances of 6.5 Å or less between the Cβ atoms of substrate alanines and the Cβ atoms of Y153/V154 (**p1**), or the Cγ atoms of V202 (**p2**), or the Cγ atom of H230 (RKH). (**C**) Top view of interactions of pore-1 loops with substrate (class 4). (**D**) Top view of interactions of pore-2 loops with substrate (class 4). (**E**) Interaction of RKH loops in the class-3 structure with the globular portion of the substrate above the channel. Inset – representative RKH-loop density (class 4, subunit C) and positions of R228, K229, and H230. (**F**) Mutation of RKH motifs in each subunit of a ClpX[ΔN] hexamer inhibits degradation of Arc-st11-ssrA. (**G**) Effects of RKH mutations on ATP hydrolysis. (**H**) Mutating two RKH motifs in a single chain pseudohexamer to AAA increases $K_M$ for steady-state degradation of [CP7]GFP-ssrA.

The online version of this article includes the following figure supplement(s) for figure 4:

**Figure supplement 1.** Substrate interactions with ClpX.

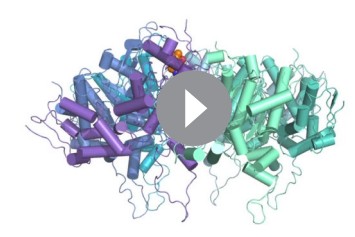

**Video 2.** Interaction of substrate with ClpX^ΔN. Substrate in the axial channel of the class-4 ClpX structure adopts an extended conformation and is contacted by the pore-1, pore-2, and RKH loops of ClpX, which are arranged into a spiral and pack against each other.

https://elifesciences.org/articles/52774#video2

and hydrolyzed ATP slightly faster (*Figure 4G–H*). Thus, the RKH loops play important roles in substrate recognition and in regulating rates of ATP hydrolysis.

## Nucleotide binding and motor conformations

We observed density for ATPγS in five subunits of each ClpX^ΔN hexamer (*Figure 5A*; *Supplementary file 1*-Table S2). In the sixth subunit, bound nucleotide density fit best as ADP, which is a 10–15% contaminant in ATPγS preparations, but was slightly less convincing than ATPγS density in other subunits (*Figure 5B*; *Supplementary file 1*-Table S2), raising the possibility of averaging of multiple nucleotide-binding states. 'ADP-bound' subunits had poorly structured channel loops, made fewer contacts with substrate and with neighboring subunits,

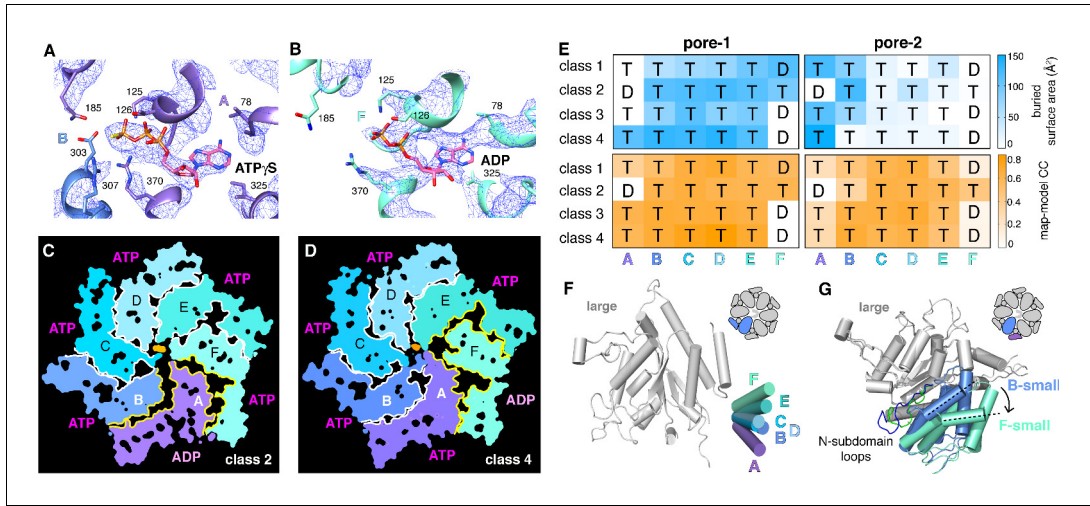

**Figure 5.** Nucleotide binding and subunit interactions in ClpX^ΔN hexamers. (**A**) Density for ATPγS in subunit A of the class-4 hexamer. The positions of ATP-binding/hydrolysis motifs are shown in cartoon and stick representation. (**B**) Weaker nucleotide density built as ADP in subunit F of the class-4 hexamer. (**C, D**) Slices through the density maps of class-2 and class-4 hexamers show that the ADP-bound subunit (A or F, respectively) makes fewer neighbor contacts than ATP-bound subunits (yellow). (**E**) Contacts between substrate and pore-1 loops (left) or pore-2 loops (right) were characterized by buried surface area (top) or model-to-map correlation (bottom). T represents an ATPγS-bound subunit; D represents an ADP-bound subunit. (**F**) Superimposition of the large AAA+ domains of six subunits from a hexamer (class 4) showing variation in the angle between small and large domains of the same subunit. One of the large AAA+ domains is light grey, and the small AAA+ domains are in different shades of purple, blue and green. (**G**) Superimposition of two large AAA+ domains from a class-4 hexamer, illustrating different types of packing against the neighboring small AAA+ domain. The large domains of subunits A and C are light grey; the small domains of subunit F and B are cornflower blue and mint green. 65-111 subdomain loops are colored green (subunit F) and blue (subunit B). The dashed lines and arrow show an ~30° rotation at the interface. The F/A interface is the seam.

The online version of this article includes the following figure supplement(s) for figure 5:

**Figure supplement 1.** Nucleotide-binding pocket and mutations.

**Figure supplement 2.** Invariant and flexible portions of ClpX^ΔN.

and were offset from the axis of the hexamer by ~2.5 Å (*Figure 5C–E*; *Supplementary file 1*-Table S3).

Side-chain contacts with nucleotide included ClpX residues Val[78]-Ile[79] (box-II), Lys[125]-Thr[126]-Leu[127] (Walker A), Asp[184]-Gln[185] (Walker B with E185Q mutation), Arg[307] (arginine finger), and Arg[370] (sensor-II) (*Figure 5A and B*, *Figure 5—figure supplement 1A*). V78A/I79A, E185Q, and R370K mutations eliminate ATP hydrolysis or weaken nucleotide binding (*Joshi et al., 2004*; *Hersch et al., 2005*; *Martin et al., 2005*; *Stinson et al., 2013*). We found that K125M, T126A, L127A, D184A, and R307A mutations also severely inhibited ATP hydrolysis (*Figure 5—figure supplement 1B*). ClpX does not have a traditional sensor-I residue, which in many AAA+ enzymes positions a water molecule for hydrolysis of ATP bound to the same subunit (*Erzberger and Berger, 2006*). For ATPγS bound in subunits A-E of our structures, the side chain of Glu[303] was close to the γ-thiophosphate, at a sensor-I-like position (*Figure 5A*). We found that an E303A mutation severely inhibited ATP hydrolysis, whereas E303Q was partially defective (*Figure 5—figure supplement 1B–C*). Based on these results, we refer to Glu[303] as a sensor-I* element. As Glu[303] is in the same helix as Arg[307], the arginine-finger residue, these residues could coordinate structurally to activate ATP hydrolysis in a neighboring subunit.

A short hinge connects the large and small AAA+ domain of each ClpX[ΔN] subunit. In all of our structures, the conformations of the small domains were very similar to each other, as were those of the large domains after residues 65–114, which comprise a subdomain, and the pore-1, pore-2, RKH, and IGF loops were removed (*Figure 5—figure supplement 2A*). The relative orientations of the large and small AAA+ domains were similar across all four structural classes for subunits at equivalent spiral positions, but conformational changes in the hinge resulted in different orientations of the large and small domains at many positions in the spiral for each hexamer (*Figure 5F*). Changes in hinge length or deletion of one hinge largely eliminate ClpX function (*Glynn et al., 2012*; *Bell et al., 2018*). Thus, the conformational changes associated with a power stroke likely arise from changes in hinge conformations, whereas movements of different large-domain loops or the 65–114 subdomain mediate ring closure and asymmetric contacts with ClpP and substrate.

ATP hydrolysis requires proper positioning of the box-II, Walker-A, and Walker-B elements in the large AAA+ domain of one ClpX subunit, the sensor-II arginine from the small AAA+ domain of the same subunit, and the arginine-finger/sensor-I* element from the large domain of the clockwise subunit (viewed from the top of the ring). These structural features depend on how the small AAA+ domain of each subunit packs against the large AAA+ domain of its clockwise neighbor, which was similar for units A/B, B/C, C/D, D/E, and E/F, suggesting that the associated ATP-binding sites are hydrolytically active. By contrast, the structure of the FA interface was different as a consequence of changes in rotation of the 65–114 subdomain in subunit A relative to the rest of the large AAA+ domain and changes in a loop that contains the sensor-II arginine in subunit F (*Figure 5G*). These changes resulted in disengagement of the Arg-finger and sensor-I* side chains in subunit A from the nucleotide bound to subunit F (*Figure 5B*; *Supplementary file 1*-Table S4). Thus, in each of our structures, the nucleotide-binding site in subunit F appears to be catalytically inactive.

## Discussion

### ClpX interactions with ClpP and substrates

Our cryo-EM structures provide snapshots of the binding of *E. coli* ClpX[ΔN] to ClpP, to protein substrates, and to nucleotides. In each of our structures, the six subunits of the ClpX ring hexamer are arranged in a shallow spiral. Slightly altered orientations of the large and small AAA+ domains in each ClpX subunit allow the hexameric ring to remain topologically closed with each large domain contacting the small domain of one neighbor. These structural results are consistent with biochemical experiments that show that ClpX is fully functional when all of its subunit-subunit interfaces are covalently crosslinked (*Glynn et al., 2012*) and support a model in which the architectural changes in the spiral that drive a power stroke result from changes in the conformations of the hinges connecting the large and small AAA+ domains of each subunit. Our structures show that relatively flexible interactions between IGF loops of ClpX and binding pockets on ClpP heptamers allow docking of these symmetry-mismatched partners. Although IGF-ClpP contacts are highly dynamic in solution (*Amor et al., 2016*), they were well defined in our structures and revealed essential interactions.

Protein substrates were observed in the axial channel of ClpX in all structures and above the channel in two structures. The ClpX pore-1 and pore-2 loops were responsible for most substrate contacts in the channel, with a periodicity of two substrate residues per ClpX subunit. The RKH loops of ClpX, which we found play critical roles in substrate recognition and control of ATP-hydrolysis rates, also contacted substrate near the top and above the channel. Substrate contacts near the top of the channel are most important in determining substrate grip during unfolding (*Bell et al., 2019*), and substrate-ClpX contacts were tightest in this region in our structures. Finally, we observed ATPγS bound to five of the six ClpX subunits with ADP likely bound to the sixth subunit, and suggest how ClpX functions without a traditional sensor-I residue.

In contrast to our present cryo-EM structures, crystal structures of *E. coli* ClpX$^{\Delta N}$ pseudohexamers (*Glynn et al., 2009*; *Stinson et al., 2013*) do not form spirals, bind only four nucleotides, and have conformations incompatible with ClpP and substrate binding. These observations highlight the fact that low-energy conformations observed by any structural method may or may not be relevant to biological function. A recent cryo-EM structure of *Listeria monocytogenes* ClpXP (*Gatsogiannis et al., 2019*), which was crosslinked to stabilize complexes, differs from our *E. coli* structures in having full-length ClpX bound to just the ClpP2 ring of heteromeric ClpP1/ClpP2, in ClpP2 having a narrower pore than ClpP in our structures, in not containing protein substrate, and in displaying head-to-head dimerization interactions between the N-terminal domains of two ClpX hexamers. Full-length *E. coli* ClpX is more aggregation prone than the ΔN variant, and negative-stain EM analysis of full-length *E. coli* ClpX bound to ClpP fails to support the importance of specific head-to-head dimerization between ClpX hexamers (*Grimaud et al., 1998*; *Ortega et al., 2000*; *Ortega et al., 2002*). Recent cryo-EM structures of *Neisseria meningitidis* ClpX$^{\Delta N}$/ClpP (*Ripstein et al., 2020*) are generally similar to those presented here but show fewer contacts between substrate and the pore-2 or RKH loops.

## Proteolytic motors from different AAA+ clades have similar structures

AAA+ protease motors belong to either the classic or HCLR clades (*Erzberger and Berger, 2006*). For HCLR-clade members ClpX and Lon, the spiral hexamer architectures and pore-1-loop interactions with substrate in the axial channel are similar to those for the classic-clade YME1/FtsH and the proteasomal Rpt$_{1-6}$/PAN motors (*Puchades et al., 2017*; *de la Peña et al., 2018*; *Dong et al., 2019*; *Majumder et al., 2019*; *Shin et al., 2019*). Thus, from a structural perspective, it is reasonable to suggest that motors from different clades may operate by a common fundamental mechanism.

A sequential translocation model has been proposed for Yme1 and the Rpt$_{1-6}$ ring of the 26S proteasome, based on placing distinct cryo-EM structures with different nucleotide and substrate-engagement states in a defined kinetic pathway, (*Puchades et al., 2017*; *de la Peña et al., 2018*; *Dong et al., 2019*). In this model, ATP hydrolysis in the fifth spiral subunit (subunit E in ClpX) drives a power stroke that moves each subunit from its previous location to a position offset by one subunit in the clockwise direction, generating a two-residue translocation step (*Figure 6A*). Vps4 was proposed to remodel proteins by a similar mechanism, but with ATP hydrolysis in the D subunit driving sequential progression of subunits into new spiral positions (*Monroe et al., 2017*). We refer to this general translocation model as SC/2R (Sequential Clockwise/2-Residue Step). Similar models have been proposed for Lon, for PAN/20S, and for the AAA+ protein-remodeling machines ClpB/Hsp104, and CDC48/p97 (*Gates et al., 2017*; *Ripstein et al., 2017*; *Su et al., 2017*; *Sun et al., 2017*; *Zehr et al., 2017*; *Yu et al., 2018*; *Cooney et al., 2019*; *Majumder et al., 2019*; *Rizo et al., 2019*; *Twomey et al., 2019*; *Shin et al., 2019*).

## Does ClpX function by an SC/2R mechanism?

The ClpX structures reported here resemble those of Yme1 and Rpt$_{1-6}$ in the spiral architecture of the hexamer, in the positions of subunits that contain a nucleoside triphosphate, in the interaction of successive pore-1 loops with two-residue segments of substrate, and in the patterns of substrate engaged and disengaged pore-1 and pore-2 loops in the ring (*Puchades et al., 2017*; *de la Peña et al., 2018*; *Dong et al., 2019*). Thus, based on structural considerations alone, ClpXP might also be expected to operate by an SC/2R mechanism. We cannot exclude this possibility. However, as discussed below, predictions of the SC/2R-model conflict with the results of multiple ClpXP experiments.

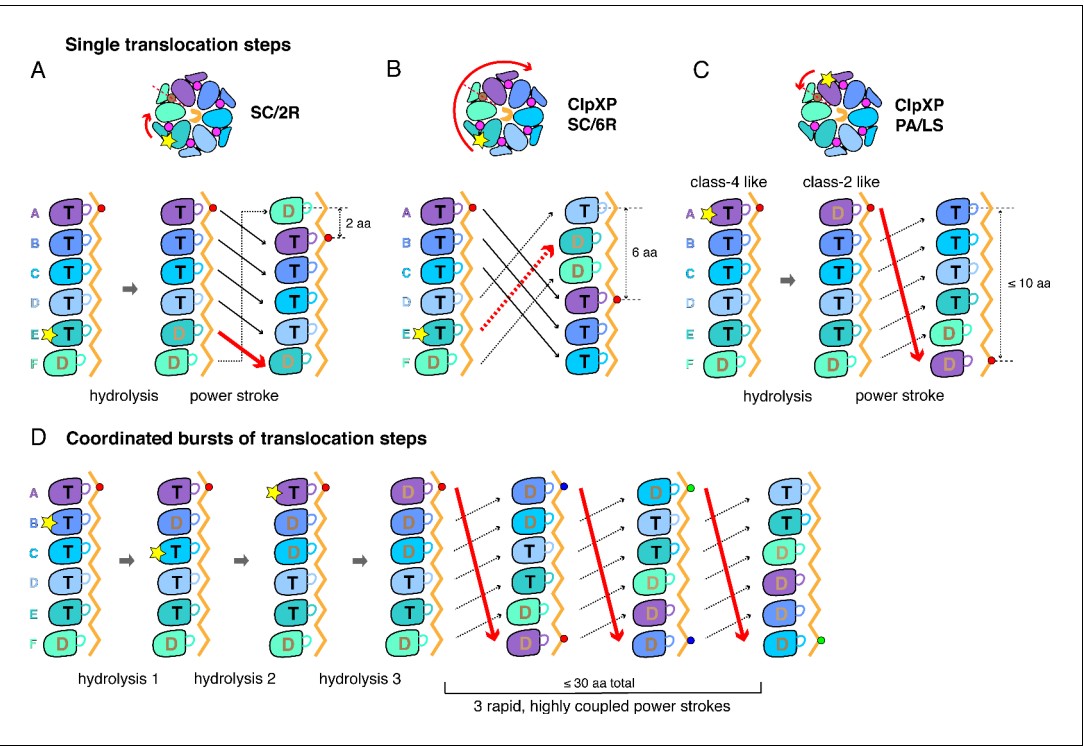

**Figure 6.** Single-step and burst translocation models. In all panels, T represents ATP-bound subunits, and D represents subunits containing ADP and possibly inorganic phosphate. (**A**) SC/2R translocation model proposed for the Yme1 protease and 26S proteasome (***Puchades et al., 2017***; ***de la Peña et al., 2018***; ***Dong et al., 2019***). Only subunit E hydrolyzes ATP (depicted by a star). Hydrolysis and/or product release results in a two-residue translocation step, in which the top five subunits in the spiral move down one position in the clockwise direction and the bottom subunit moves up to the top position. (**B**) SC/6R translocation model. During a six-residue translocation step, subunits A, B, and C each move down in the spiral to positions D, E, and F, respectively, dragging substrate with them; at the same time, subunits D, E and F each move up to positions A, B, and C. Subunit displacement is in the clockwise direction. (**C**) PA/LS model in which ATP hydrolysis in subunit A results in a single translocation step of up to 2.5 nm in length as a consequence of anti-clockwise movement of this subunit to position F at the bottom of the spiral. At the same time, subunits BCDEF move up to positions ABCDE. (**D**) One variation of a PA/LS model resulting in a burst of three long translocation steps. Initial probabilistic ATP hydrolysis in subunits D and then B creates strain in the spiral, which is released in burst of fast steps upon ATP hydrolysis in subunit A. In the general PA/LS model, the initial ATP-hydrolysis event can occur with different probabilities in subunits A-E.

The online version of this article includes the following figure supplement(s) for figure 6:

**Figure supplement 1.** ATP-hydrolysis rates.

One major issue involves the length of translocation steps. The SC/2R model predicts a two-residue step. By contrast, the basic translocation step of ClpXP measured in optical-trapping experiments appears to be ~6 residues, with some steps as much as fourfold larger (***Aubin-Tam et al., 2011***; ***Maillard et al., 2011***; ***Sen et al., 2013***; ***Cordova et al., 2014***; ***Iosefson et al., 2015b***; ***Rodriguez-Aliaga et al., 2016***; ***Olivares et al., 2017***). These single-molecule experiments measure average length changes in the portion of a protein substrate outside of the axial channel as a consequence of ClpXP unfolding (increased distance) or translocation (decreased distance). For example, ClpXP unfolding of filamin-A domains results in 14–19 nm distance increases (***Aubin-Tam et al., 2011***), with the distance being greater at higher force in accord with the worm-like chain model (***Schlierf et al., 2007***; ***Ferrer et al., 2008***). As a filamin-A domain consists of ~100 residues, 14–19 nm corresponds to 0.14–0.19 nm/res or ~5–7 res/nm. Thus, the unfolded polypeptide outside the channel is partially compact, as a fully extended conformation would be 0.34 nm/res (~3 res/nm), similar to the extension of the substrate in the axial channel of our cryo-EM structures. During ClpXP translocation in the optical trap, distance decreases in steps of approximately 1, 2, 3, or 4 nm. These

results indicate that the basic translocation step is ~6 residues, with longer steps presumably representing kinetic bursts of this fundamental step.

A second important issue involves translocation kinetics. The sensitivity of optical trapping precludes direct identification and quantification of translocation steps as short as two residues, and thus it might be argued that ~6 residue steps consist of three unresolved SC/2R sub-steps. Consideration of rates, however, makes this possibility unlikely. For example, the time from the beginning to the end of each translocation step is less than 0.1 s in the optical trap. Under similar conditions, the steady-state rate of ClpX$^{\Delta N}$/ClpP ATP hydrolysis is 3.6 ± 0.1 s$^{-1}$ (*Figure 6—figure supplement 1*), corresponding to a time constant of 0.28 s. The SC/2R model predicts that each sub-step would require an independent ATP hydrolysis event, including ADP/P$_i$ dissociation and ATP rebinding, and thus that an average of ~0.8 s would be required to take three sub-steps. This kinetic problem becomes worse for translocation bursts that move 12 to 24 residues in less than 0.1 s. Thus, ClpXP translocation occurs at rates ~ 8 to~32 fold faster than predicted by the SC/2R mechanism and the experimentally determined rate of ATP hydrolysis. Another issue sometimes raised is that ClpX step sizes could be longer under tension in optical-trapping experiments, or that ClpXP uses a different translocation mechanism under these conditions. We consider both possibilities unlikely as the distribution of translocation step lengths shows little dependence on trap force. Moreover, ClpXP unfolding and translocation rates measured by optical trapping are similar to those determined in single-molecule fluorescence assays (*Shin et al., 2009*).

A third issue involves the prediction of the SC/2R model that ATP hydrolysis occurs at only one position in the ClpX spiral during translocation. If this prediction were correct, then single-chain ClpX variants containing subunits that cannot hydrolyze ATP should stall when an ATPase inactive subunit moves into the hydrolysis position. However, ClpX$^{\Delta N}$ hexamers with just two ATPase-active subunits support ClpP degradation at ~30% of the wild-type rate and with the same thermodynamic efficiency as wild-type ClpXP (*Martin et al., 2005*). Could thermal motions move an ATPase-inactive subunit out of the hydrolysis position in the ClpX spiral and an active subunit into it, allowing continued function? This possibility seems unlikely, as the variant with just two ATPase active subunits translocates substrates against force in optical-trap experiments (*Cordova et al., 2014*). As we discuss below, non-SC/2R models that permit probabilistic hydrolysis by subunits at different positions in the spiral could explain how stalling is prevented when multiple ClpX subunits are hydrolytically inactive and could also provide a mechanism for rapid bursts of translocation steps.

## Structure-based translocation models for ClpXP

To account for the experimentally determined translocation properties of ClpXP, any model needs to explain: (1) how structural changes in the spiral result in a fundamental translocation step of ~6 residues; (2) how kinetic bursts could generate very fast translocation of as many as ~24 residues without requiring multiple ADP-dissociation and ATP-rebinding events; and (3) how the ClpX motor functions efficiently without strict requirements for ATP hydrolysis in any specific subunit in the spiral.

Modification of the SC/2R model could, in principle, address the step-size issue. For example, upon ATP hydrolysis in a unique subunit in the spiral, a clockwise three-subunit shift could

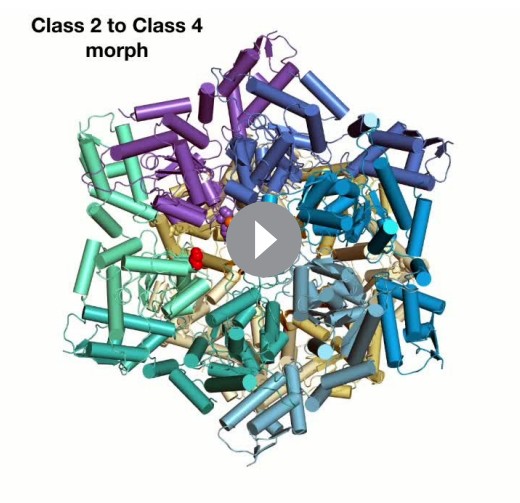

**Video 3.** Transition between class-2 and class-4 structures of ClpX$^{\Delta N}$/ClpP. A conformational morph between the class-2 and class-4 structures of ClpXP is shown. To generate this morph, ClpX subunits in the class-2 structure were renamed from ABCDEF to FABCDE and aligned to the class-4 ClpXP structure in two steps to maximize overlap between the ClpP portions of the two structures. The ADP-bound ClpX subunit (colored aquamarine) moves most dramatically, from the top to the bottom of the ClpX spiral. In the ADP-bound subunit, residues of the pore-1 loop move ~1.6–1.9 nm.
https://elifesciences.org/articles/52774#video3

generate a six-residue power stroke in a SC/6R model (*Figure 6B*). Potential issues with this mechanism include that substrate contacts with multiple subunits would need to break during each translocation step, the structural features that would drive a three-subunit rearrangement are unclear, and this SC/6R model fails to account for kinetic bursts resulting in longer translocation steps.

To account for single ~6 residue translocation steps and longer kinetic bursts, we propose a <u>P</u>robabilistic <u>A</u>nti-clockwise <u>L</u>ong-<u>S</u>tep (PA/LS) model (*Figure 6C and D*), which has two major components. First, a power stroke in the anti-clockwise direction moves the top subunit in the ClpX spiral to the bottom position, thereby translocating substrate by ~1.7 nm. A conformational change of this type is captured in a morph in which subunit A in class two is aligned with subunit F in class 4 (*Video 3*); a similar transition has been considered for Vps4 (*Su et al., 2017*). We assume that ATP-hydrolysis in subunit A of ClpX powers this step, although this is not critical for the general model. Repeating this cycle sequentially would account for ~6 residue steps, assuming that the polypeptide is translocated in an extended conformation, but not for bursts of longer steps. Thus, we also posit that initial ATP hydrolysis could occur with some probability in ClpX subunits B through E, with these events causing strain in the spiral that is released in a burst of power strokes when ATP in subunit A is eventually hydrolyzed or sufficient strain accumulates. By allowing coupling of multiple ATP-hydrolysis events prior to translocation, this model obviates the need to release ADP and rebind ATP prior to each step, allowing fast kinetic bursts. Moreover, from a structural perspective, five of the six nucleotide-binding sites in our ClpX structures have similar geometries in at least one structural class, and thus could plausibly hydrolyze ATP.

The PA/LS model implies that subunit A can bind substrate at the top of the spiral and then drag or push it to the bottom. In this regard, we note that the pore-1 loop of subunit-A contacts substrate in our class-4 structure, with these contacts buttressed by interactions with nearby RKH loops. This model would also require transient breaking of contacts between the substrate and other spiral subunits during the power stroke, but this could be done in a zipper-like fashion to reduce kinetic barriers.

By allowing ATP hydrolysis at multiple spiral positions, probabilistic variations of the SC/2R and SC/6R models (PC/2R and PC/6R) can also be formulated, which would allow the possibility of kinetic bursts. We note that probabilistic models do not preclude neighboring subunits in the ring from firing sequentially, but rather exclude any mechanistic requirement for sequential action. It has been argued that probabilistic or stochastic models imply that subunits in the AAA+ ring must be completely independent (*Smith et al., 2011*), but this is not true. Communication between ClpX subunits and cooperative ATP hydrolysis has been documented (*Hersch et al., 2005*; *Martin et al., 2005*) and is compatible with probabilistic models.

It is not clear how many structural changes are involved in the complete reaction cycle for ClpXP or related AAA+ ATPases. For example, crystal structures of ClpX reveal rotations between the large and small AAA+ domains so large that nucleotide cannot bind some subunits (*Glynn et al., 2009*; *Stinson et al., 2013*). Although subunits of this type are not present in current cryo-EM structures, they might represent transient functional conformations, as crosslinks engineered to trap these hexamer conformations form in solution and prevent ClpXP degradation but not ATP hydrolysis (*Stinson et al., 2013*; *Stinson et al., 2015*). Additional substrate-bound conformations of ClpXP probably also await discovery, as it is unclear from current structures how ClpX initially recognizes substrate degrons. Moreover, ClpXP degrades disulfide-bonded and knotted proteins in reactions that require simultaneous translocation of two or more polypeptides (*Burton et al., 2001*; *Bolon et al., 2004*; *San Martín et al., 2017*; *Sivertsson et al., 2019*). In our cryo-EM structures, a single polypeptide strand fills the axial channel at its narrowest point, which would have to expand to accommodate multiple strands during translocation. Vps4 accommodates two polypeptides in its axial channel without major distortion of its hexameric spiral (*Han et al., 2019*). A similar mechanism might allow ClpX to translocate two polypeptide chains, although it is difficult to imagine accommodation of three or more chains. Finally, we note that the SC/2R mechanism requires substrates to be translocated in an extended conformation, whereas ClpXP translocates poly-proline tracts that are unable to form this structure (*Barkow et al., 2009*).

## Other AAA+ unfolding/remodeling machines and SC/2R mechanisms

ClpAP takes ~1 nm and ~2 nm residue translocation steps (*Olivares et al., 2014*; *Olivares et al., 2017*), suggesting that it can also operate by a non-SC/2R mechanism, but we are unaware of

experiments that establish the translocation step size for other AAA+ protein unfolding or remodeling enzymes. Like ClpX, some of these enzymes do not stall if an ATPase-inactive subunit occupies the hydrolysis position. In the $Rpt_{1-6}$ hexamer of the yeast 26S proteasome, for example, ATPase-defective $Rpt_3$, $Rpt_4$, or $Rpt_6$ subunits cause virtually complete loss of degradation activity, whereas ATPase-defective $Rpt_1$, $Rpt_2$, and $Rpt_5$ subunits do not (*Beckwith et al., 2013*). In the alternating Yta10/Yta12 hetero-hexamer (a Yme1 homolog), a Walker-B mutation in Yta12 prevents ATP hydrolysis in Yta10, as expected for tight subunit-subunit coupling in a strictly sequential mechanism, but a Walker-B mutation in Yta10 does not prevent ATP hydrolysis in Yta12 (*Augustin et al., 2009*). Given the strong similarities between the structures and substrate interactions of a large number of AAA+ unfolding and remodeling machines and the fact that the SC/2R mechanism does not account for ClpX and ClpA experimental results, it will be important to continue to use structural and functional studies to investigate the molecular mechanisms by which these AAA+ machines function.

# Materials and methods

## Key resources table

| Reagent type (species) or resource | Designation | Source or reference | Identifiers | Additional information |
|---|---|---|---|---|
| Gene (*Escherichia coli*) | clpX | *E. coli* (strain K12) EXPASY | UniProtKB- P0A6H1 | |
| Gene (*Escherichia coli*) | clpP | *E. coli* (strain K12) EXPASY | UniProtKB- P0A6G7 | |
| Recombinant DNA reagent | PACYC-ClpX$^{\Delta N}$(E185Q)$_6$-TEV-cHis$_6$ (plasmid) | This paper, Material and methods | | ClpX expression, can be obtained from the Sauer lab |
| Recombinant DNA reagent | pT7ClpP-TEV-cHis$_6$ (plasmid) | (*Stinson et al., 2013*) | | WTClpP expression, can be obtained from the Sauer lab |
| Recombinant DNA reagent | pT7-ClpPR192K-TEV-cHis6 (plasmid) | This paper, Material and methods | | ClpP mutant, can be obtained from the Sauer lab |
| Recombinant DNA reagent | pT7-ClpX$^{\Delta N}$-AAA (plasmid) | This paper, Material and methods | | ClpX RKH loop mutant, can be obtained from the Sauer lab |
| Recombinant DNA reagent | pT7-ClpX$^{\Delta N}$-AKH (plasmid) | This paper, Material and methods | | ClpX RKH loop mutant, can be obtained from the Sauer lab |
| Recombinant DNA reagent | pT7-ClpX$^{\Delta N}$-KHR (plasmid) | This paper, Material and methods | | ClpX RKH loop mutant, can be obtained from the Sauer lab |
| Recombinant DNA reagent | pT7-ClpX$^{\Delta N}$-KRH (plasmid) | This paper, Material and methods | | ClpX RKH loop mutant, can be obtained from the Sauer lab |
| Recombinant DNA reagent | PACYC-ClpX$^{\Delta N}$-(RKH$_2$-AAA)$_2$ (plasmid) | This paper, Material and methods | | ClpX RKH loop mutant, can be obtained from the Sauer lab |
| Recombinant DNA reagent | pET4b-Arc-st11-ssrA (plasmid) | (*Bell et al., 2018*) | | Expresses Arc substrate for degradation assay, can be obtained from the Sauer lab |
| Recombinant DNA reagent | pT7-$^{cp7}$GFP-ssrA (plasmid) | (*Nager et al., 2011*) | | Expresses fluorescent substrate for degradation assay, can be obtained from the Sauer lab |
| Strain, strain background (*Escherichia coli*) | ER2566 | NEB | 1312 | Chemical competent cells |

*Continued on next page*

*Continued*

| Reagent type (species) or resource | Designation | Source or reference | Identifiers | Additional information |
|---|---|---|---|---|
| Chemical compound, drug | ADEP-2B | (*Schmitz et al., 2014*) | | Small molecule antibiotics binds ClpP. |
| Software, algorithm | Relion | (*Kimanius et al., 2016*) | RRID:SCR_016274 | EM reconstruction software |
| Software, algorithm | UCSF Chimera | (*Pettersen et al., 2004*) | RRID:SCR_004097 | Molecular Visualization Software |
| Software, algorithm | UCSF ChimeraX | (*Goddard et al., 2018*) | RRID:SCR_015872 | Molecular Visualization Software |
| Software, algorithm | Phenix | (*Adams et al., 2010*) | RRID:SCR_014224 | Protein Model Building Software |
| Software, algorithm | Molprobity | (*Williams et al., 2018*) | RRID:SCR_014226 | Protein Model Evaluation Software |
| Software, algorithm | Caver | (*Pavelka et al., 2016*) | | Molecular Visualization Software |
| Software, algorithm | PyMOL | Schrödinger, LLC. | RRID:SCR_000305 | Molecular Visualization Software |
| Software, algorithm | Coot | (*Emsley and Cowtan, 2004*) | RRID:SCR_014222 | Protein Model Building Software |
| Software, algorithm | Ctffind | (*Rohou and Grigorieff, 2015*) | RRID:SCR_016732 | EM Image Analysis Software |
| Software, algorithm | PISA | 'Protein interfaces, surfaces and assemblies' service PISA at the European Bioinformatics Institute. (http://www.ebi.ac.uk/pdbe/prot_int/pistart.html) | RRID:SCR_015749 | Protein Model Analyzing Software |

## Protein expression and purification

The single-chain ClpX variant used for microscopy contained six copies of *E. coli* E185Q ClpX$^{\Delta N}$ (residues 62–424), with neighboring units connected by six-residue linkers of variable composition, and a C-terminal TEV cleavage site and His$_6$ tag. The *E. coli* ClpP variant consisted of an N-terminal propeptide, which was removed auto-proteolytically during expression/purification, residues 1–193 of mature ClpP, and the TEV-His$_6$ epitope. Both proteins were expressed separately in *E. coli* strain ER2566 and purified as described (*Stinson et al., 2013*). After purification, TEV protease was used to remove the His$_6$ tags. TEV protease and uncleaved proteins were removed by Ni$^{2+}$-NTA affinity chromatography, and purified proteins were flash frozen in storage buffer (20 mM HEPES, pH 7.5, 300 mM KCl, 0.5 mM EDTA, 10% glycerol) and stored at –80°C.

## Sample preparation, data acquisition, and image processing

To assemble complexes, the ClpX$^{\Delta N}$ pseudohexamer and ClpP$_{14}$ were diluted into EM buffer (20 mM HEPES, pH 7.5, 100 mM KCl, 25 mM MgCl$_2$, 5 mM ATPγS) to final concentrations of 4 µM and 1.8 µM, respectively. After 5 min at 25°C, 3 µL of the mixture was applied to glow discharged R1.2/1.3 400 mesh grids (Quantifoil). Grids were blotted with filter paper 494 (VWR) and plunged into liquid ethane using a Cryoplunge-3 system (Gatan). Electron micrographs were collected using a Talos Arctica with a Gatan K2-Summit direct electron detector in super-resolution mode. High-resolution movies were recorded at a magnification of 36000X (0.58 Å pixel size). Each movie was composed of fifty frames (200 ms per frame) and a total dose of ~58 e$^-$/Å$^2$ per movie. The final dataset

consisted of 3657 movies recorded in two separate sessions. Frames in each movie were 2X binned, aligned, gain-corrected, and dose-weighted using Motioncor2 (*Zheng et al., 2017*), to generate a single micrograph. The contrast transfer function (CTF) was estimated using CTFFIND4 (*Rohou and Grigorieff, 2015*). Unless noted, Relion 2.0 (*Kimanius et al., 2016*) was used for 2D/3D classification and refinement.

We first attempted to construct a density map of doubly capped ClpX-ClpP-ClpX using standard protocols. 1.4 million doubly capped particles were automatically picked and filtered by 2D classification. 443,717 'good' particles were selected for 3D map reconstruction. To generate an initial model for 3D refinement, the crystal structures of a ClpX$^{\Delta N}$ hexamer (PDB 3HWS; *Glynn et al., 2009*) and a ClpP tetradecamer (PDB code 3MT6; *Li et al., 2010*) were merged in PyMOL and low-pass filtered to 40 Å. 3D refinement with $C_2$ symmetry yielded a map with a resolution of ~4 Å, but the quality of this map was poor and interpretation of secondary structure elements was impossible. Using $C_1$ or $C_7$ symmetry did not improve the map.

To minimize problems caused by the symmetry mismatch between ClpX and ClpP, we treated ClpX and ClpP separately before the last step of refinement. For the ClpP reconstruction, we applied a soft circular mask, including ClpP and the tips of the ClpX IGF loops, to the 3D reference and 2D images, respectively. Because we observed predominantly side views of ClpXP (perpendicular to the axial channel; *Figure 1—figure supplement 1A–C*), this simple masking of the particle images allowed us to remove most of the ClpX density and to focus particle alignment on ClpP. Starting from a low-pass filtered ClpP structure as an initial reference, 3D refinement with $D_7$ symmetry yielded a 3.2 Å resolution map with clear secondary structure and side-chain features. For the ClpX reconstruction, we extracted ClpX sub-particles from both ends of ClpP based on the previously determined ClpP alignment using a python script and IMOD (*Mastronarde and Held, 2017*). We prepared an original and a ClpP-signal-subtracted ClpX particle stack. To remove misaligned particles, the ClpX stacks were 2D classified without alignment. Particles from 2D classes that showed clear secondary structure were used in subsequent 3D-classification and reconstruction steps.

Clean ClpX sub-particles were 3D classified (K = 6) and refined, resulting in four distinct ClpX classes with resolutions ranging from 3.9 to 4.2 Å. To recover the ClpX$^{\Delta N}$-ClpP interface, we re-extracted ClpXP sub-particles using a larger box that included the cis ClpP ring. Alignment and classification of each ClpX$^{\Delta N}$ sub-particle was transferred to corresponding ClpX$^{\Delta N}$/ClpP sub-particle and four classes were refined with local alignment optimization, resulting in four ClpX$^{\Delta N}$/ClpP maps (resolutions 4.0 to 4.3 Å).

To test the robustness of this workflow, we performed 3D classification of ClpX sub-particles multiple times using K = 4 or K = 8. The quality of maps suffered slightly, but the overall structures of four predominant classes of ClpX hexamers remained unchanged.

## Model building and refinement

A ClpP tetradecamer (PDB 3MT6; *Li et al., 2010*) was docked into the $D_7$ map using Chimera's 'fit to map' function (*Pettersen et al., 2004*). For ClpX$^{\Delta N}$, six copies of the large and small AAA+ domains (PDB 3HWS; *Glynn et al., 2009*) were docked into the map sequentially and refined for three iterations in Chimera. Real-space refinement of docked ClpP and ClpX was performed using PHENIX (*Adams et al., 2010*), and model building was performed using Coot (*Emsley and Cowtan, 2004*). ChimeraX (*Cordova et al., 2014*), Chimera (*Pettersen et al., 2004*), and PyMOL were used to create figures and movies.

## Biochemical assays

Assays were conducted at 37°C in PD buffer (25 mM HEPES-KOH, pH 7.5, 5 mM MgCl$_2$, 200 mM KCl, 10% glycerol). Experiments were performed in triplicate and reported values are averages ± SD. Degradation of $^{CP7}$GFP-ssrA (15 μM) by ClpX$^{\Delta N}$ (0.3 μM hexamer) and ClpP$_{14}$ (0.9 μM) was assayed in the presence of 5 mM ATP and was monitored by loss of fluorescent signal (excitation 467 nm; emission 511 nm). For RKH loop mutants with reduced substrate affinity, degradation was measured at high concentrations of fluorescent substrate, with excitation at an off-peak wavelength (excitation 420 nm; emission 511 nm). Degradation of fluorescein-labeled Arc-st11-ssrA by ClpX$^{\Delta N}$ (0.3 μM hexamer) and ClpP$_{14}$ (0.9 μM) was assayed as described (*Bell et al., 2018*). $K_M$ and $V_{max}$ were

determined by fitting the average values of replicates to a hyperbolic equation. ATP hydrolysis rates were measured using a coupled-NADH oxidation assay as described (*Martin et al., 2005*), using ClpX$^{\Delta N}$ (0.3 µM hexamer), with or without ClpP$_{14}$ (0.9 µM), and 5 mM ATP. Activation of decapeptide cleavage by ClpP or variants was performed as described (*Lee et al., 2010a*), using ClpP$_{14}$ (50 nM), RseA decapeptide (15 µM), ATP (5 mM) and a regeneration system, and either ClpX$^{\Delta N}$ (0.5 µM hexamer) or ADEB-2B (100 µM), which was a generous gift from J. Sello (Brown). Pull-down experiments were performed in 25 mM HEPES-KOH (pH 7.5), 5 mM MgCl$_2$, 150 mM KCl, 20 mM imidazole, 10% glycerol, 500 µM dithiothreitol and 2 mM ADP or ATPγS. 40 µL of a mixture of ClpX$^{\Delta N}$ C169S (1 µM hexamer concentration) and ClpP (1 µM 14-mer concentration) were mixed, incubated for 10 min at 30°C, and then added to 20 µL of Ni$^{2+}$-NTA resin (Thermo Scientific) equilibrated in the same buffer. Binding reactions were incubated for 15 min at room temperature with rotation, centrifuged for 1 min at 9400 x g, and the supernatant was discarded. Reactions were washed, centrifuged, and the supernatant discarded three times. Bound protein was eluted with 40 µL of buffer supplemented with 300 mM imidazole for 15 min with rotation. Reactions were then centrifuged again and the eluant collected. Input and elution samples for each reaction were resolved by SDS-PAGE on a 10% Bis-Tris/MES gel run at 150V and visualized by staining with Coomassie Brilliant Blue R250. Results were validated in three independent replicates.

## Acknowledgements

Supported by NIH grant GM-101988 (RTS) and the Howard Hughes Medical Institute (TA Baker; SC Harrison). We thank S Glynn, M Lang, H Manning, A Olivares, and K Schmitz for helpful discussions, J Sello for providing ADEP-2B, and C Xu and K Song at the Electron Microscopy Facility at the University of Massachusetts Medical School for advice and data collection. The authors declare no competing financial interests.

## Additional information

### Funding

| Funder | Grant reference number | Author |
|---|---|---|
| National Institutes of Health | GM-101988 | Robert T Sauer |
| Howard Hughes Medical Institute | | Tania A Baker Stephen C Harrison |
| National Institutes of Health | 5T32GM-007287 | Tristan A Bell |

The funders had no role in study design, data collection and interpretation, or the decision to submit the work for publication.

### Author contributions

Xue Fei, Conceptualization, Formal analysis, Investigation, Methodology; Tristan A Bell, Simon Jenni, Investigation, Methodology; Benjamin M Stinson, Investigation; Tania A Baker, Supervision; Stephen C Harrison, Supervision, Methodology; Robert T Sauer, Supervision, Funding acquisition, Investigation

### Author ORCIDs

Tristan A Bell http://orcid.org/0000-0002-3668-8412
Tania A Baker http://orcid.org/0000-0002-0737-3411
Stephen C Harrison http://orcid.org/0000-0001-7215-9393
Robert T Sauer https://orcid.org/0000-0002-1719-5399

### Decision letter and Author response

Decision letter https://doi.org/10.7554/eLife.52774.sa1
Author response https://doi.org/10.7554/eLife.52774.sa2

## Additional files

### Supplementary files
• Supplementary file 1. contains four Tables with information about the structures.

• Transparent reporting form

### Data availability
PDB files for the structures determined here have been deposited in the PDB under accession codes 6PPE, 6PP8, 6PP7, 6PP6, 6PP5, 6POS, 6POD, 6PO3, and 6PO1.

The following datasets were generated:

| Author(s) | Year | Dataset title | Dataset URL | Database and Identifier |
|---|---|---|---|---|
| Fei X, Jenni S, Harrison SC, Sauer RT | 2020 | D7 structure of ClpP and ClpX IGF loops | http://www.rcsb.org/structure/6PPE | RCSB Protein Data Bank, 6PPE |
| Fei X, Jenni S, Harrison SC, Sauer RT | 2020 | ClpX class-1 structure | http://www.rcsb.org/structure/6PP8 | RCSB Protein Data Bank, 6PP8 |
| Fei X, Jenni S, Harrison SC, Sauer RT | 2020 | ClpX class-2 structure | http://www.rcsb.org/structure/6PP7 | RCSB Protein Data Bank, 6PP7 |
| Fei X, Jenni S, Harrison SC, Sauer RT | 2020 | ClpX class-3 structure | http://www.rcsb.org/structure/6PP6 | RCSB Protein Data Bank, 6PP6 |
| Fei X, Jenni S, Harrison SC, Sauer RT | 2020 | ClpX class-4 structure | http://www.rcsb.org/structure/6PP5 | RCSB Protein Data Bank, 6PP5 |
| Fei X, Jenni S, Harrison SC, Sauer RT | 2020 | ClpXP class-1 structure | http://www.rcsb.org/structure/6POS | RCSB Protein Data Bank, 6POS |
| Fei X, Jenni S, Harrison SC, Sauer RT | 2020 | ClpXP class-2 structure | http://www.rcsb.org/structure/6POD | RCSB Protein Data Bank, 6POD |
| Fei X, Jenni S, Harrison SC, Sauer RT | 2020 | ClpXP class-3 structure | http://www.rcsb.org/structure/6PO3 | RCSB Protein Data Bank, 6PO3 |
| Fei X, Jenni S, Harrison SC, Sauer RT | 2020 | ClpXP class-4 structure | http://www.rcsb.org/structure/6PO1 | RCSB Protein Data Bank, 6PO1 |

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
