## [Decision Letter]

**Acceptance summary:**

The translocation mechanism of processive AAA+ unfoldases has been of significant interest and debate for some time. Although a consensus 'rotary cycling' view has emerged for how ATPase activity is coupled to substrate movement through a hexameric motor pore, the present work proposes a significantly different mechanism, invoking a probabilistic, rather than sequential, firing order. These findings are certain to generate discussion in field.

**Decision letter after peer review:**

Thank you for submitting your article "Structures of the ATP-fueled ClpXP proteolytic machine bound to protein substrate" for consideration by *eLife*. Your article has been reviewed by three peer reviewers, and the evaluation has been overseen by a Reviewing Editor and Cynthia Wolberger as the Senior Editor. The following individuals involved in review of your submission have agreed to reveal their identity: Andreas Matouschek (Reviewer #1); Christopher P Hill (Reviewer #2).

The reviewers have discussed the reviews with one another and the Reviewing Editor has drafted this decision to help you prepare a revised submission.

Summary:

The present paper reports the first structure of ClpXP, with ClpX in what appears to be an active, translocating conformation with nucleotide and bound substrate. Bacterial ClpXP is an ATP-fueled chambered protease, akin to the 26S proteasome in eukaryotic cells. The hexameric AAA+ unfoldase (ClpX) recognizes, unfolds, and threads SsrA-tagged substrates into the tetradecameric barrel composed of two heptameric ClpP rings, where the ssrA-substrate is degraded. ClpX is an established model system that has served as an exemplar for AAA unfoldases in general.

The asymmetric spiral conformations of ClpX presented in the paper have been largely anticipated because they resemble a number of other AAA unfoldase-substrate complexes that have been reported over the last 2-3 years. Although confirmatory, the demonstration that ClpX can adopt such a conformation is significant, because the ATPase had been seen previously to adopt different conformations in earlier crystal structures and a recent cryo-EM model. Other valuable insights from this work include showing how ClpX stabilizes the open pore conformation of ClpP, how ClpX binds to ClpP, and that contacts between pore-loop 1 and substrate can be mediated by both Tyr153 and the flanking Val154. The latter finding argues that it is not the aromatic residue but the hydrophobicity of pore loop 1 that is important for substrate interaction.

Essential revisions:

1) The proposed mechanistic model for ClpX conflicts with sequential ATPase cycling models favored by most (but not all) other papers that have imaged AAA+ unfoldases in complex with substrates or substrate-mimics. Although there is some doubt as to whether the proposed probabilistic mechanism is likely to be correct, there was consensus that it will stimulate debate by its challenge of what is rapidly becoming the orthodox mechanism for AAA unfoldases. The reviewers also note that the probabilistic model comports with certain biochemical observations in the ClpX system, such as the ability of ATPase to translocate substrate with only a small subset of functional catalytic sites. Nonetheless, there is a sense that the authors may be too quick to fully dismiss sequential mechanisms based on their data. In particular, the interaction of five ClpX subunits with substrate via loop-1 and loop-2 would seem more consistent with a sequential or "conveyer belt" model. Additional discussion of this issue is warranted.

2) Pertaining to (1), the authors acknowledge that their model requires a conformation that has not yet been visualized. Specifically, from the text: "In any PA/LS model, subunit A would need to bind substrate and drag it to the bottom of the spiral, at least transiently breaking contacts with other subunits. Whether these contacts would be physically broken or simply released during the power stroke is currently unclear, as not all conformational states in the ClpX reaction cycle are likely to be known at this point."

A major concern is with envisioning a conformation in which the one transitioning subunit (which in the current structures does not contact substrate) binds the peptide while the other subunits (which here bind cooperatively to an extended stretch of the substrate) release substrate. Why/how would the transitioning subunit bind the substrate more tightly than the cooperative binding of the other five subunits? The authors' conclusion that relevant conformations have not yet been visualized is possible, but this seems like a major weakness relative to the sequential model. Please address this major concern.

3) Not all descriptions of the sequential model have suggested that ATP hydrolysis occurs at the subunit E active site. Indeed, Monroe et al., 2017, which was the first of these structures to be reported, proposed that hydrolysis occurs in the subunit D active site (at the interface with subunit E). Please address this concern.

4) There is a question regarding the statement: "In the optical trap, however, step size represents the average distance that unfolded polypeptide outside of the axial channel moves between successive translocation steps, which can be converted into amino-acid residues using the wormlike-chain model (Bustamante et al., 1994). Because the unfolded substrate outside the channel is in a partially compact conformation at the forces used in these experiments, ~1 nm corresponds to a translocation step of 5-8 residues, which appears inconsistent with the two-residue step predicted by the SC/2R model."

The citation (Bustamante, 1994) given here is for the wormlike model as applied to DNA. Please cite or otherwise justify the statement that unfolded substrate outside the channel will be partially compact and in a worm-like state with 5-8 residues per nm under the conditions of the experiments. This may be evident from the citations a little earlier in the text, but it would be helpful to have an explicit indication of the basis for this assertion at this point in the Discussion.

5) The critique of the sequential model in accommodating substrates that are more complex than a single polypeptide strand raises some questions. Specifically, the text states: "In our cryo-EM structures, a single polypeptide strand fills the axial channel, and thus structures must exist in which the channel expands to accommodate multiple strands during translocation."

The ability of the asymmetric spiral structure and the sequential model to accommodate complex substrates without expanding the channel is indicated by Han et al., 2019, which reports structures of Vps4 in which two substrate polypeptide chains bind without distorting the AAA unfoldases structure. The authors should overlay their structure on this Vps4 model to visualize if a similar configuration of two bound polypeptide chains is plausible for ClpX.

6) The present work is preceded by a recent publication of *monocytogenes* ClpXP (Gatsogiannis et al., 2019). Surprisingly, Gatsogiannis et al. reported the existence of unusual head-to-head ClpXP dimers that were prevalent in their cryoEM sample but are not observed in the present work. Whether head-to-head dimers are artifactual or physiological is currently unclear. Furthermore, contrary to the present work, singly capped complexes were predominant if not exclusively observed. It seems appropriate to compare and comment on the previously published structures and to discern why such head-to-head dimers were not observed in the present study.

---

## [Author Response]

Essential revisions:1) The proposed mechanistic model for ClpX conflicts with sequential ATPase cycling models favored by most (but not all) other papers that have imaged AAA+ unfoldases in complex with substrates or substrate-mimics. Although there is some doubt as to whether the proposed probabilistic mechanism is likely to be correct, there was consensus that it will stimulate debate by its challenge of what is rapidly becoming the orthodox mechanism for AAA unfoldases. The reviewers also note that the probabilistic model comports with certain biochemical observations in the ClpX system, such as the ability of ATPase to translocate substrate with only a small subset of functional catalytic sites. Nonetheless, there is a sense that the authors may be too quick to fully dismiss sequential mechanisms based on their data. In particular, the interaction of five ClpX subunits with substrate via loop-1 and loop-2 would seem more consistent with a sequential or "conveyer belt" model. Additional discussion of this issue is warranted.

In the revised manuscript, we explicitly state that the SC/2R sequential model is consistent with our cryo-EM structures and cannot be excluded. We continue to say, however, that this 'orthodox' model makes multiple predictions that disagree with experimental results. In each case of disagreement (translocation-step size; rates of translocation; and whether hexamers with multiple ATPase-inactive subunits should be active in protein unfolding/translocation), we discuss arguments advanced by advocates of the orthodox model concerning ways in which the experimental data could be misleading and why we disagree. In the revised text, we note that rates of ClpXP substrate unfolding and translocation determined by optical trapping are similar to those determined by single-molecule fluorescence experiments. We also discuss the fact that there are few, if any, direct experimental tests of the orthodox model. Although we obviously come down on the side of heterodox models, we feel that we have now provided sufficient information for readers to decide for themselves.

2) Pertaining to (1), the authors acknowledge that their model requires a conformation that has not yet been visualized. Specifically, from the text: "In any PA/LS model, subunit A would need to bind substrate and drag it to the bottom of the spiral, at least transiently breaking contacts with other subunits. Whether these contacts would be physically broken or simply released during the power stroke is currently unclear, as not all conformational states in the ClpX reaction cycle are likely to be known at this point."A major concern is with envisioning a conformation in which the one transitioning subunit (which in the current structures does not contact substrate) binds the peptide while the other subunits (which here bind cooperatively to an extended stretch of the substrate) release substrate. Why/how would the transitioning subunit bind the substrate more tightly than the cooperative binding of the other five subunits? The authors' conclusion that relevant conformations have not yet been visualized is possible, but this seems like a major weakness relative to the sequential model. Please address this major concern.

First, Tyr^153^ and Val^154^ in the pore-1 loop of subunit A do contact substrate in our class-4 structure. We emphasize this fact in the revised manuscript. Second, as we note in the revised manuscript, in the proposed PA/LS mechanism substrate contacts with different ClpX subunits during a power stroke may be broken in a zipper-like fashion rather than a cooperative all-or-none fashion. Finally, structure alone does not allow calculation of the free energies required to break observed interactions. Thus, whether it would be easy or hard to break substrate contacts with 4/5 pore-1 loops isn't clear, and would depend both on factors such as differential solvation, entropy, and the structures of transition states (for kinetics) and other ground states (for equilibria).

3) Not all descriptions of the sequential model have suggested that ATP hydrolysis occurs at the subunit E active site. Indeed, Monroe et al., 2017, which was the first of these structures to be reported, proposed that hydrolysis occurs in the subunit D active site (at the interface with subunit E). Please address this concern.

We thank the reviewers for drawing attention to this oversight and now cite the subunit-D hydrolysis model proposed by Monroe et al., 2017, in the revised Discussion.

4) There is a question regarding the statement: "In the optical trap, however, step size represents the average distance that unfolded polypeptide outside of the axial channel moves between successive translocation steps, which can be converted into amino-acid residues using the wormlike-chain model (Bustamante et al., 1994). Because the unfolded substrate outside the channel is in a partially compact conformation at the forces used in these experiments, ~1 nm corresponds to a translocation step of 5-8 residues, which appears inconsistent with the two-residue step predicted by the SC/2R model."The citation (Bustamante, 1994) given here is for the wormlike model as applied to DNA. Please cite or otherwise justify the statement that unfolded substrate outside the channel will be partially compact and in a worm-like state with 5-8 residues per nm under the conditions of the experiments. This may be evident from the citations a little earlier in the text, but it would be helpful to have an explicit indication of the basis for this assertion at this point in the Discussion.

We removed the Bustamante reference and replaced it with Schlierf et al., 2007, and Ferrer et al., 2008 references in which experiments show that the wormlike-chain (WLC) model is applicable to denatured polypeptides under force. More importantly, we discuss optical-trapping experiments (Aubin-Tam et al., 2011) in which ClpXP unfolding of filamin-A domains results in 14-19 nm distance changes. The magnitude of the change depends on force in accord with the WLC model (greater extension at higher force; see Author response image 1).

As the filamin-A domain consists of ~100 residues, simple division results in values of 0.14-0.19 nm/res, corresponding to ~5-7 res/nm. In a fully extended conformation, by contrast, the extension would be 0.34 nm/res, corresponding to ~3 res/nm. Thus, the polypeptide outside the ClpX channel is partially compact compared to a fully extended state and substrate within the channel.

5) The critique of the sequential model in accommodating substrates that are more complex than a single polypeptide strand raises some questions. Specifically, the text states: "In our cryo-EM structures, a single polypeptide strand fills the axial channel, and thus structures must exist in which the channel expands to accommodate multiple strands during translocation."The ability of the asymmetric spiral structure and the sequential model to accommodate complex substrates without expanding the channel is indicated by Han et al., 2019, which reports structures of Vps4 in which two substrate polypeptide chains bind without distorting the AAA unfoldases structure. The authors should overlay their structure on this Vps4 model to visualize if a similar configuration of two bound polypeptide chains is plausible for ClpX.

We did not intend the fact that ClpXP can degrade knotted and crosslinked proteins as a critique of the SC/2R sequential model but to indicate that there are likely to be ClpXP structures bound to substrate that have not yet been observed. At it’s narrowest point, the radius of the axial channel of ClpX in our class-1 structure is ~2 Å (Figure 3B), which is too narrow to accommodate a second polypeptide. In the revised manuscript, we acknowledge that a conformation similar to that described by Han et al. could allow accommodation of a second polypeptide, but we are not convinced that such a structure could accommodate three or four polypeptides (as is required for degradation of knotted substrates or those with internal disulfide bonds). We also note in the revised text that ClpXP can translocate tracts of polyproline, which cannot form the extended substrate structure that we and others observe in the axial channel and that current structures yield no insight into initial recognition of degrons like the ssrA tag by ClpXP. Again, these observation suggest that there are more functional conformations of ClpXP to discover. We believe that it is highly unlikely that all functionally relevant conformations of ClpXP or any other AAA+ machine are currently known.

6) The present work is preceded by a recent publication of Listeria monocytogenes ClpXP (Gatsogiannis et al., 2019). Surprisingly, Gatsogiannis et al. reported the existence of unusual head-to-head ClpXP dimers that were prevalent in their cryoEM sample but are not observed in the present work. Whether head-to-head dimers are artifactual or physiological is currently unclear. Furthermore, contrary to the present work, singly capped complexes were predominant if not exclusively observed. It seems appropriate to compare and comment on the previously published structures and to discern why such head-to-head dimers were not observed in the present study.

The head-to-head dimers in the *Listeria monocytogenes* ClpXP structure are mediated by interactions between cysteine-rich N-terminal domains of ClpX (which were deleted and thus not present in our structures) and were only observed in samples cross-linked prior to cryo-EM. Negative-stain EM analysis of full-length *E. coli* ClpX bound to ClpP does not substantiate the claim of a specific head-to-head dimerization interaction between ClpX Ndomains (Grimaud et al., 1998; Ortega et al., 2000; 2002). The *Listeria monocytogenes* complexes contained one heptamer of ClpP1 and one heptamer of ClpP2. ClpX only bound to the ClpP2 ring, presumably because ClpP1 lacks appropriate binding determinants. In our structures, both ClpP rings are identical and the ratio of complexes singly capped or doubly capped by ClpX depends largely on mixing concentrations. As requested, we now discuss these issues in the revised Discussion. We also mention cryo-EM structures of Neisseria meningitidis ClpX^∆N^/ClpP that are generally similar to ours but contain fewer interactions between substrate and the pore-2 or RKK loops.